

# Phenotypic diversity and provenance variation of *Cupressus funebris*: a case study in the Sichuan Basin, China

Wang Yan[1],*, Yongqi Xiang[1],*, Mei Gao[1], Ruoyu Deng[1], Yan Sun[1], Renping Wan[1], Xianyi Pan[1], Wanzhen Li[1] and Yu Zhong[1,2]

[1] College of Forestry, Sichuan Agricultural University, Chengdu, Sichuan Province, China
[2] National Forestry and Grassland Administration Key Laboratory of Forest Resources Conservation and Ecological Safety on the Upper Reaches of the Yangtze River & Forestry Ecological Engineering in the Upper Reaches of the Yangtze River Key Laboratory of Sichuan Province, Chengdu, China
* These authors contributed equally to this work.

Corresponding author
Yu Zhong, zhongyu315@163.com

## ABSTRACT

**Background:** The species *Cupressus funebris* holds substantial ecological value and economic potential, particularly in the realms of rehabilitating barren mountain landscapes and advancing urban greening endeavors. However, inadequate attention has been given to research endeavors exploring the genetic diversity and morphological characteristics of *Cupressus funebris*, a deficiency that could potentially hinder its development, utilization, and conservation of genetic resources.
**Methods:** To rectify the shortage of existing basic morphological data, a morphological analysis was conducted in this study on 180 *Cupressus funebris* germplasm resources sourced from five provenances. Key traits explored included growth characteristics, overall morphology, branch/leaf features, and seed traits. This will facilitate the evaluation of genetic diversity in *Cupressus funebris*.
**Results:** The findings reveal a considerable level of phenotypic variation (PVC of 16.9%) and genetic diversity (1.97 index) in *Cupressus funebris* germplasm resources. The phenotypic differentiation is observed to be 48% between provenances and 52% within provenances, primarily attributed to variation originating from individual provenances. Based on these morphological characteristics, the germplasm resources have been categorized into four distinct groups: Ecological Restoration Planting, Secondary Reserve Forest, Urban Greening, and Timber Forest. Interestingly, the pattern of variation observed within these groups is irregular, exhibiting no significant correlation with their respective provenances. Furthermore, conifer trees sharing similar growth characteristics tend to display comparable patterns of random variation, suggesting potential underlying genetic mechanisms. This study significantly enriches the phenotypic dataset within the genetic variation research of *Cupressus funebris*, facilitating development and utilization for ecological construction, timber breeding, and horticultural greening.

## INTRODUCTION

Genetic diversity, the abundance of genetic information within a species, reflects its level of endangerment, environmental adaptability, and evolutionary potential (*Frankham, 2012*; *Hoban et al., 2021*; *Li et al., 2022a*), serving as a cornerstone guiding plant conservation and breeding efforts (*Jones et al., 2006*). As research techniques advance, scholars are increasingly utilizing contemporary molecular technologies to explore plant genetic diversity (*Plomion et al., 2014*; *Motalebipour et al., 2016*; *Liu et al., 2022a*; *Qing et al., 2022*), leading to substantial progress in unraveling the genetic basis and evolutionary history of diverse plant species (*Lu et al., 1996*; *Wolf, Lindell & Backström, 2010*). This achievement is deserving of recognition. However, during genetic variation and adaptive evolution in plants, the initial response manifested by plants towards their external environment is invariably observed at the phenotypic level. This is evident in the observable differences in plant traits, commonly referred to as phenotypic variation, which can be discerned without the use of specialized equipment.

The existence of phenotypic variation is essential for the emergence of phenotypic diversity, a key component of genetic diversity in morphological studies (*Chowdhury, Jana & Schroeder, 2000*). Intraspecific genetic variation and the phenotypic plasticity of plants contribute to phenotypic variation (*Wu, 1998*; *Sugai et al., 2023*). The adaptive evolution of plants in response to variable environments results in the divergence and formation of distinct morphological groups at the population level, which facilitates the inheritance of genetic variation to subsequent generations (*González-Martínez, Krutovsky & Neale, 2006*). This illustrates that phenotypic changes can, to some extent, reflect altered plant genetic information (*Kuijper & Johnstone, 2021*). Therefore, phenotypic variability was employed as a fundamental criterion for plant breeding selection. Meanwhile, the use of phenotypic evaluation represents a classical approach in genetic information research for assessing genetic diversity (*Zhang et al., 2015*). In comparison to alternative approaches for studying genetic diversity, while molecular techniques such as genomics could provide faster and more efficient information on the genetic variation of a species, our understanding of genetic information is always limited without the objective evaluation of phenotypic data (*Beaton et al., 2022*). Consequently, phenotyping remains a pivotal methodology for the advancement and utilization of plant resources.

The term "provenance" pertains to the geographic origin of seeds or other propagation materials. This concept holds significant importance in the genetic improvement of trees (*Míguez-Soto & Fernández-López, 2015*; *Alizoti, Aravanopoulos & Ioannidis, 2019*). Trees of the same species, when grown in diverse natural environmental conditions for prolonged periods, exhibit genetic traits and geographical variations that mirror local environmental conditions (*Cortan, Nonic & Sijacic-Nikolic, 2019*). As a result, these germplasm materials effectively represent the growth patterns of natural geographic populations and demonstrate greater genetic diversity compared to a single geographic population (*Barzdajn, Kowalkowski & Chmura, 2016*). Under uniform growth conditions, these genetic variations manifest themselves morphologically, resulting in differences in

growth characteristics among species such as *Alnus cremastogyne* (*Zheng et al., 2023*) and variations in fruit size in *Prunus avium* (*Popovic et al., 2020*).

The *Cupressus funebris* (*C. funebris*), an indigenous species of *Cupressus* in China, exhibits remarkable adaptability to its environment. The species possesses well-developed lateral roots, enabling it to thrive even on infertile and shallow soils. This renders it a suitable candidate for pioneering afforestation and ecological restoration of barren mountainous regions (*Wang et al., 2021*; *Li et al., 2022b*). Furthermore, its graceful and upright posture has made *C. funebris* a popular choice for landscaping in China. Consequently, a significant wild seed collection project was initiated in Sichuan Province, China, during the 1970s and 1980s with the objective of afforesting barren mountains. The wild germplasm materials, which were investigated and collected during this project, exhibit significant phenotypic variations and abundant intraspecific genetic diversity. For instance, they display a range of cone sizes and diverse seed shapes, spanning from near-rhomboid to circular. Notably, some of the ancient trees investigated have trunk diameters exceeding 1 m, demonstrating extraordinary growth ability. Furthermore, these *C. funebris* exhibit distinct growth morphologies; some varieties boast slender, pendulous twigs reminiscent of weeping willow (*Salix babylonica*), while others are characterized by shorter, bushy twigs. In 2004, the government of Sichuan Province, China (*Sichuan, 2011*), designated *C. funebris* as a crucial tree species for the development of industrial raw material forests, emphasizing its significance for the cultivation of rare trees. Consequently, it is of utmost importance to conduct relevant assessments of genetic diversity in *C. funebris*. In the field of *C. funebris* research, the majority of studies have concentrated on the testing of the antibacterial properties of essential oil and the renovation of inefficient plantation forests (*Romeo et al., 2008*; *Lohani et al., 2015*; *Xie et al., 2016*; *Wang et al., 2021*; *He et al., 2022*; *Wen et al., 2022*; *Yuan & Hao, 2023*). However, there has been a paucity of research on genetic diversity in *C. funebris*. Only a few studies have employed SSR markers in parental breeding, assessed genetic information through microsatellite markers, and conducted straightforward comparisons of growth traits (*Lu et al., 2014*; *Yang et al., 2016*; *Yang et al., 2023a*). Moreover, there is a lack of pertinent morphological research, which may hinder the genetic improvement and utilization development of *C. funebris*.

Consequently, this study evaluates the genetic diversity of 180 *C. funebris* trees across 19 traits within a seed orchard, utilizing five provenances and 60 half-sibling families, with the support of the national superior *C. funebris* germplasm base located in Santai County, Mianyang City, Sichuan Province. The objective of this study is to elucidate phenotypic variations and the degree of differentiation among *C. funebris* individuals, thereby strengthening the research framework on the genetic variation of this species. Furthermore, the study aims to provide valuable insights into the genetic management of *C. funebris* germplasm resources and contribute to the development and utilization of greening tree species.

## MATERIALS AND METHODS

### Study area and materials

The research site, which was established in 1987, is situated in Santai County, Sichuan Province. It serves as the seed orchard of the National *Cupressus funebris* Seed Base. The site is located at a longitude of 105°14' east and latitude of 31°22' north in the north-central part of the Sichuan Basin (Fig. S1). The average elevation of the site is 501 m, and it experiences a subtropical humid monsoon climate. This is characterized by an average annual rainfall of 882.2 mm and a temperature of 16.7 °C. The soil has an approximate thickness of 50 cm, with a pH value ranging from 7.5 to 8.0 and a thin detritus layer. The seed orchard was divided into six large zones, and the geographic conditions were essentially consistent across the six zones. Furthermore, uniform management practices were implemented across the six zones. After a period of natural selection, plant spacing now ranges from 3 m × 3 m to 3 m × 5 m. In this experiment, a total of 60 half-sibling families from five distinct provenances within the seed orchard were selected as the research material. For each family, three individual plants were randomly selected for subsequent phenotypic measurements. The families were sourced from five locations: Bazhong City (BZ), Guangyuan City (GY), Nanchong City (NC), Santai County (ST) of Mianyang City, and Nanjiang County (NJ) of Bazhong City. This resulted in a total of 180 *C. funebris* germplasm resources (Tables S1 and S8).

### Trait measurement and calculation

#### Growth traits

A survey and sampling were conducted during the peak fruit maturation season in autumn, specifically mid-October 2020. The dimensions of *C. funebris* trees were recorded in the seed orchard, including tree height, diameter at breast height, crown width, and branch height. Tree height and branch height were quantified using a laser ultrasound survey instrument (Vertex Laser Geo), while diameter at breast height and crown width were measured with a tape measure. The width of the crown was calculated as the mean of the east-west and north-south measurements. The crown height was calculated based on the following formula:

$$CH = H - BH \tag{1}$$

where $CH$ is the crown height, $H$ is the tree height, and $BH$ is the branch height. The wood volume was calculated based on the following formula (Yang et al., 2023a):

$$V = 0.000057173591 \times DBH^{1.8813305} \times H^{0.99568845} \tag{2}$$

where $V$ is wood volume, $DBH$ is the diameter at breast height, and $H$ is the height of trees.

#### Leaf traits

The annual branches on the standard branch were collected by cutting the standard branch from the east, west, south, and north directions. Subsequently, the annual branches were scanned utilizing a BenQ A3 flatbed scanner, with a ruler included in the scanning frame to establish a benchmark for subsequent image calibration. Following this, the lengths and
angles of 10 annual branches were analyzed with the aid of ImageJ software (version 1.52a). In addition, three leaf angles were randomly selected for measurement for each branch.

### Cone and seed traits

A diverse set of mature cones was collected from standard branches, and from each tree, 15 cones were randomly selected. These cones were measured using vernier calipers to determine their longitudinal and transverse diameters, and the number of cone scales was recorded. Following this, the cones underwent a drying and threshing process in an oven maintained at a temperature of 40 °C. Afterward, a total of 100 seeds were selected from the dried cones for weighing to ascertain their collective 100-grain weight. The collected seeds were then scanned using a BenQ A3 flatbed scanner to create digital images, utilizing the same calibration method as for the branches. From these scanned images, 20 seeds were chosen for precise length and width measurements using the ImageJ software (version 1.52a). Since the cone is very close to spherical, its volume is calculated using the formula for spherical volume.

## Statistical analyses

Microsoft Excel 2019 was used for basic data statistics in this study, while Minitab Statistical Software 2021 was utilized for the two-factor nested ANOVA (with the family factor nested within the provenance factor), and IBM SPSS Statistics 27 (IBM, Armonk, NY, USA) was employed for the correlation analysis and principal component analysis. Additionally, the Duncan multiple comparison method was applied to assess the means and standard deviations of phenotypic traits. Finally, the corresponding images were generated using Origin 2021 and R 4.3.2. The linear model of variance analysis according to the following statistical model:

$$Y_{ijk} = \mu + P_i + A(P)_{i(j)} + \varepsilon_{ijk} \tag{3}$$

where $Y_{ijk}$ is an individual plant observation; $\mu$ is the overall mean; $P_i$ is the effect of among provenances; $A(P)_{i(j)}$ is the effect of individual plant within provenances; and $\varepsilon_{ijk}$ is the random error. The calculation formula of phenotypic coefficient of variation, Shannon-Wiener genetic diversity index and phenotypic differentiation coefficient (*Zhang et al., 2015*) is as follows:

$$CV = \frac{SD}{\overline{X}} \times 100\% \tag{4}$$

$$H = -\sum_{i=1}^{s} p_i \times \ln p_i \tag{5}$$

$$V_{st} = \frac{\sigma_{t/s}^2}{\left(\sigma_{t/s}^2 + \sigma_s^2\right)} \tag{6}$$

where *CV*, *H* and *Vst* are the coefficient of variation, Shannon–Wiener genetic diversity

index and phenotypic differentiation coefficient, respectively. The *SD* is the standard deviation of the mean value of a trait, $\bar{X}$ is the mean value of trait, $p_i$ is the probability of occurrence at level $i$ of a trait, $\sigma^2_{t/s}$ is the variance component among provenances, and $\sigma^2_s$ is the variance component within provenances.

## RESULTS

### Phenotypic trait and diversity

The fundamental characteristics of the 19 traits of *C. funebris* are presented in Table S2. It is notable that the annual branch angle exhibited the highest standard deviation (8.64) with a mean value of 60.8 degrees. Subsequently, the leaf angle exhibited the second-highest standard deviation (5.24) with a mean value of 40.8 degrees. Conversely, the hundred-grain weight exhibited the lowest standard deviation (0.06) and had a mean value of 0.19 g. With regard to the coefficients of variation, the hundred-grain weight exhibited the highest value (32.6%), followed by the cone volume (31.3%). On the contrary, the lowest coefficient of variation was 10% for cone transverse diameter. The Shannon-Wiener genetic diversity index for the 19 traits in *C. funebris* exhibited a range of values between 1.65 and 2.08. The minimum value was observed for tree height, while the maximum value was observed for crown width and leaf angle (Fig. S2).

A comparison of the variation among five provenances revealed that GY exhibited the highest coefficient of variation (15.5%), while BZ displayed the lowest (12.1%). With regard to the genetic diversity index, the NJ provenance exhibited the highest value (1.96), while NC demonstrated the lowest (1.70). The average coefficients of variation and genetic diversity index for the 180 germplasm resources were 16.9% and 1.97, respectively, indicating high phenotypic variation and genetic diversity (see Tables S6 and S7 for detailed data).

### Variance analysis among and within provenances

The analysis of variance (ANOVA) revealed significant differences ($p < 0.05$) among the five provenances (BZ, GY, NC, NJ, and ST) and highly significant ($p < 0.01$) differences within the provenances for *C. funebris* traits, with the exception of cone scales number (Table S3, and Tables S9–S13). Figure S3 presents the results of the Duncan multiple comparisons performed on various traits among the distinct provenances. For growth traits, the NC provenance exhibited the highest average tree height (14.1 m, as high as the ST provenance), branch height (5.6 m, as high as the ST provenance) and crown height (8.6 m). The provenance ST exhibited the largest values for diameter at breast height, wood volume, and crown width (31.7 cm, 0.535 m$^3$, and 8.5 m, respectively). Conversely, the BZ provenance showed the smallest tree height (11.1 m), wood volume (0.263 m$^3$), and branch height (3.6 m), which were significantly smaller than those of the other provenances ($p < 0.05$). The NC provenance had the smallest crown width (6 m), while the GY provenance had the smallest crown height (7.2 m).

In terms of branching and leafing traits, the GY provenance exhibited the largest annual branch length (31.9 cm), significantly greater than the other provenances ($p < 0.05$). The NC provenance had the largest values for both annual branch angle (65.6°) and leaf angle

(45°), significantly greater than those observed in the other provenances ($p < 0.05$) (Figs. S3J–S3L).

With regard to cone and seed traits, the ST provenance exhibited a significantly larger mean value than the other provenances ($p < 0.05$), while the differences in cone and seed traits among the other provenances were relatively minor (Figs. S3M–S3S).

## Phenotypic differentiation

Figure S4 shows the structural differentiation of provenances for 19 traits in *C. funebris*. The analysis of variance components revealed that tree height represented the greatest proportion of variance among provenances (67.2%), while the number of cone scales number represented the smallest (6.3%). With regard to the within provenances variance components, the annual branch angle demonstrated the highest value (49.9%), whereas the wood volume exhibited the lowest (6.1%). In summary, the breakdown of variance components revealed that 29.8% was attributed to differences among provenances, 29% to variations within provenances, and 41.2% to random error (Figs. S4A and S4B).

Figures S4C and S4D illustrate the extent of phenotypic differentiation among and within provenances. The *Vst* for tree height was the highest among provenances (87.6%), while it was the lowest within provenances for this trait (12.4%) (Fig. S4C). Conversely, cone scales number exhibited the highest *Vst* within provenances (83.6%), accompanied by the correspondingly lowest *Vst* among provenances (16.4%) (Fig. S4D). A comparison of different trait categories revealed that the *Vst* among provenances was higher for growth traits compared to other traits. In contrast, the *Vst* within provenances for branch and leaf traits, as well as seed traits, was significantly higher than for growth traits. In summary, the *Vst* among provenances and within provenances was 48% and 52%, respectively.

## Correlation analysis of phenotypic traits

The correlation analysis revealed significant ($p < 0.05$) or highly significant ($p < 0.01$) associations among several of the 19 traits in *C. funebris* (Fig. S5). Specifically, highly significant positive correlations were observed between tree height, diameter at breast height, wood volume, crown width, and branch height ($p < 0.01$), indicating a strong synergistic growth relationship among these traits. Furthermore, crown width showed a highly significant negative correlation with annual branch angle ($p < 0.01$), while branch height exhibited a significant negative correlation with the length of annual branch ($p < 0.05$). Additionally, crown height demonstrated a significant positive correlation with the ratio of crown height to crown width (CH/CW) ($p < 0.01$) and the length of annual branch ($p < 0.05$), but a negative correlation with the ratio of tree height to crown height (H/CH) ($p < 0.01$). This suggests that as the crown height and crown width of *C. funebris* increased, the length of the annual branch became longer, while the angle of annual branch growth tended to decrease. The annual branch angle exhibited a highly significant positive correlation with leaf angle ($p < 0.01$), indicating that as the angle of the branches increased, the leaves grew in a more open manner.

Significant positive correlations were observed between cone vertical diameter, cone transverse diameter, cone volume, cone scales number, seed length, and seed width, as well

as hundred-grain weight ($p < 0.01$), suggesting that larger cones were associated with larger seeds, thereby implying better seed quality. Furthermore, cone and seed traits exhibited significant positive correlations with tree height, diameter at breast height, wood volume and crown width ($p < 0.05$), indicating that individuals with superior growth traits in *C. funebris* tend to have more prominent cone and seed traits.

## Principal component analysis of phenotypic traits

To gain an understanding of the primary phenotypic traits distinguishing *C. funebris*, a principal component analysis was conducted, and the findings were presented in Fig. S6. Five principal components were extracted based on eigenvalues greater than one, collectively accounting for 79.82% of the cumulative contribution rate. This indicates that these five principal components encompass 79.82% of the original data information.

From the principal component matrix, it was observed that each principal component emphasized distinct trait information. The variance percentage of Principal Component 1 (PC1) was 32.8%, featuring relatively high loading values for cone vertical diameter, cone transverse diameter, cone volume, seed length, seed width, and hundred-grain weight, suggesting that PC1 represents information pertaining to reproductive traits. Principal Component 2 (PC2), with a variance percentage of 16.8%, had relatively high loading values for crown height, the ratio of tree height to crown width (H/CW), and crown height to crown width (CH/CW), indicating that it represents information related to crown traits. Principal Component 3 (PC3), on the other hand, exhibited a variance percentage of 13.6%, displaying high loading values for tree height, diameter at breast height, wood volume, and crown width, which implies that it represents a factor pertaining to growth traits. Moving on, Principal Component 4 (PC4) had a variance percentage of 10.9%, with prominent loading values for branch height and the ratio of tree height to crown height (H/CH), suggesting that it represents information related to branch height. Lastly, Principal Component 5 (PC5) showed a variance percentage of 5.7%, featuring high loading values for annual branch length, annual branch angle, and leaf angle, indicating that it represents information pertaining to branch and leaf morphology. Due to the differing contributions of each principal component, the trait information encompassed by each and their respective importance differed. The relative importance of trait information decreased in a sequential manner from PC1 to PC5.

The spatial distribution of 180 *C. funebris* individuals was illustrated in Fig. S6D and S6E. It was observed that the samples from different provenances exhibited a random dispersion without displaying any discernible pattern based on their provenance. This indicates that there is no significant correlation between trait information and provenance for *C. funebris*, as evidenced by the six principal components representing the trait information.

By aggregating the scores of each principal component, the variability of individual traits can be assessed. Table S4 revealed that the top five phenotypic traits, based on total score coefficients, were leaf angle (0.66), annual branch angle (0.629), tree height (0.602), crown height (0.536), and annual branch length (0.513). This indicated that *C. funebris* exhibited a notable degree of variation in these five traits. Conversely, the scores at the

bottom three positions were the ratio of tree height to crown height (−0.193), cone transverse diameter (0.022), and branch height (0.089), respectively, indicating that these particular phenotypic traits exhibited lower variation within *C. funebris*.

## Cluster analysis of germplasm resources

Following the normalization of the raw data, 180 samples of *C. funebris* germplasm resources from five provenances were subjected to clustering and analysis using the Ward. D method based on Euclidean distance, as illustrated in Fig. S7. At a height of 110, the 180 *C. funebris* germplasm resources were classified into four distinct groups. Group I comprised 68 *C. funebris* germplasm resources, which were characterized by the most prominent cone and seed traits and relatively superior growth traits. Group II comprised 21 *C. funebris* germplasm resources, which were characterized by the highest tree height and crown height, the smallest crown width, long branches, and an open leaf growth pattern. Group III consisted of 30 *C. funebris* germplasm resources, which were characterized by poorer growth traits, a better tree form ratio, long branches, and a contracted branch and leaf growth pattern, but inferior cone and seed traits. Group IV contained 61 *C. funebris* germplasm resources, which were characterized by the most superior growth traits, relatively superior cone and seed traits, but a poorer tree form ratio. The detailed mean values of phenotypic traits for each group are presented in Table S5.

## DISCUSSION

The intricate natural environment, prolonged geographical isolation, and the process of natural selection are the primary factors facilitating phenotypic variation in woody plants (*González-Martínez, Krutovsky & Neale, 2006*). *Cupressus funebris* is a species widely distributed across various climatic zones in China, commonly inhabiting hilly areas, karst mountains, and river valleys (*Lu et al., 2014*). Phenotypic variation arises among different populations due to a combination of genetic factors and environmental influences in heterogeneous growth conditions.

The coefficient of variation and the Shannon-Wiener genetic diversity index are commonly employed to assess phenotypic variation, providing a direct indication of the variability within the subject of research (*Zhang et al., 2022*). In this study, we selected 19 relatively important and intuitive traits of *C. funebris* to analyze the variability and genetic diversity among 180 individuals in the *C. funebris* seed orchard. Our results uncovered substantial phenotypic diversity in the traits of *C. funebris*, aligning with the findings of phenotypic studies on *Pinus koraiensis* germplasm resources (*Kaviriri et al., 2020*).

*Cupressus funebris*, a gymnosperm with a long evolutionary history, can trace its origins back to the Mesozoic era (*Spencer et al., 2015*; *Liu et al., 2022b*). Having adapted to environmental changes such as mountain range formations, it has gradually evolved to thrive in arid, cold, and alpine climates, developing xeromorphic characteristics like low tree forms, compact crowns, and short-scaled leaves, which persist in some modern populations (*Jiang & Wang, 1997*). With climate changes and geographical expansion, *C. funebris* has demonstrated remarkable adaptability (*Dakhil et al., 2019*), particularly in watershed areas, where some populations have evolved superior growth traits such as

larger tree forms, longer branches, and distinct scale leaves (*Kvacek, Manchester & Schorn, 2000*). This adaptive evolution has significantly contributed to the phenotypic diversity among *C. funebris* germplasm resources (*Ma et al., 2019*; *Hu et al., 2023*). The comprehensive genetic diversity index, measured at 1.97 in this study, underscores the robust adaptability of *C. funebris*. The 180 germplasm resources were sourced from mountainous regions surrounding basins and adjacent river basins, with notable differences in growth environments, including arid and high-altitude regions (*e.g.*, BZ) and areas with mild and humid climates (*e.g.*, NC). These differences in growth environments have led to notable variations in growth and seed-related traits (*Mastretta-Yanes et al., 2012*; *Sahib et al., 2022*; *Zheng et al., 2023*), as evidenced by the high coefficients of variation observed in wood volume (26.3%), cone volume (31.3%), and 100-grain weight (32.6%), respectively (Figs. S2A–S2E). In this study, the 180 germplasm resources possess diverse genetic backgrounds, stemming from various historical growth environments. Consequently, the high variability observed among these 180 germplasm resources is a concerted result of genetic background differences and local adaptive evolution. This means that some individuals exhibit typical xeromorphic characteristics, while others have undergone evolution in different traits in response to wetter growth environments (*Marks, 2007*; *Blanco-Sánchez et al., 2024*).

In comparison to the research conducted by *Yang et al. (2023a)*, the coefficients of variation for tree height, diameter at breast height, under branch height, and crown width of 9-year-old *C. funebris* ranged from 20.92% to 52.73%. These values were notably higher than the variability observed in the growth traits examined in this study, which ranged from 10.3% to 26.3%. Such discrepancies are likely attributable to differences in the age of the *C. funebris* samples. As trees during the juvenile stage exhibit more vigorous growth (*Ununger, Ekberg & Kang, 1988*), it is expected that growth traits would exhibit greater variability. Furthermore, in this stage, traits such as tree height, diameter at breast height, and crown width are particularly susceptible to environmental influences (*Santos-del-Blanco et al., 2013*), resulting in even greater variability. Additionally, phenotypic plasticity of plants is most pronounced during the early stages of growth (*Wu et al., 2021a*). As growth progresses and the tree matures, these traits tend to stabilize. Consequently, the observed variability in *C. funebris* growth traits in this study was less pronounced than that reported by *Yang et al. (2023a)* as was predictable.

After conducting ANOVA analyses, it was evident that there existed significant ($p < 0.05$) or highly significant ($p < 0.01$) differences in the majority of traits, not only among but also within the various provenances. Furthermore, a pivotal discovery was made: the intra-provenance variation was found to contribute more substantially to the overall variability compared to the variation between provenances, as showed in Fig. S4D. This observation is consistent with prior findings in phenotypic studies of *Pinus yunnanensis* (*Xu et al., 2016*). The observed phenotypic variations among distinct populations of *C. funebris* can be attributed to environmental factors, which result in morphological adaptations. This phenomenon, known as phenotypic plasticity (*Wu, 1998*), highlights the remarkable adaptability of *C. funebris* across diverse regions to their respective environments. Conversely, genetic differentiation is the primary factor
responsible for the observed phenotypic differences within comparable regions (*Hao et al., 2019*).

In this study, traits in *C. funebris* exhibited variations influenced by both natural environmental factors and genetic components. With regard to growth-related traits, the observed phenotypic variation among different provenances exceeded that within a single provenance. This indicates that both differences in the growth environment and genetic factors contribute to the variability in growth traits (Fig. S4C). In contrast, for branch, leaf, and seed traits, the phenotypic variation within a single provenance was found to significantly exceed that among different provenances. This suggests that these traits were primarily influenced by genetic variations (Fig. S4D).

Previous studies have revealed significant disparities in the underlying mechanisms through which environmental and genetic variations influence growth and reproductive traits. Environmental factors exert a greater influence on growth traits, resulting in morphological differentiation (*Puy et al., 2021*), whereas genetic factors play a more pivotal role in shaping reproductive traits (*Drosse et al., 2014*). These findings suggest that selecting superior varieties based on the morphological characteristics of their cones and seeds is an effective and reliable approach when undertaking genetic breeding improvements for forest trees.

This study expands the fundamental phenotypic dataset within the genetic variation research framework of *C. funebris*. The coefficient of variation and genetic diversity index for phenotypic traits in *C. funebris* were 16.9% and 1.97, respectively, indicating substantial diversity. These findings suggest a strong genetic basis for selective breeding in short-term processes (*Laakili et al., 2016*; *Tchokponhoué et al., 2020*). However, cultivating superior seeds in a seed orchard is a lengthy process that involves a continuous depletion of genetic variability. The ongoing selection process will inevitably lead to a reduction in the genetic foundation of *C. funebris*, resulting in a significant loss of genetic resources (*Siepielski & Benkman, 2010*). Therefore, the continuous evaluation and supplementation of genetic resources based on phenotypic characteristics, physiological dynamics, and genetic differences will be a long-term process, aimed at improving superior varieties through breeding. This represents our forthcoming work.

Correlation analysis is a fundamental method for investigating the interrelationships among diverse phenotypic traits. This study identified substantial correlations among growth traits, cone traits, and seed traits, which is consistent with earlier research (*Zheng et al., 2023*). The analysis of scale factors, including the ratio of tree height to crown height (H/CH) and the ratio of crown height to crown width (CH/CW), uncovered distinct growth patterns within *C. funebris*. Notably, the branch height exhibited a significant negative correlation with crown height and CH/CW ($p < 0.01$), indicating that trees with broader crowns undergo more robust natural pruning.

Furthermore, the significant negative correlation between H/CH and CH/CW ($p < 0.01$) suggests that trees with a lower ratio of tree height to crown height tend to exhibit less lateral growth. This finding indicates a preference for vertical growth in *C. funebris* within the seed orchard setting. This trend may be linked to the high density of seedling planting during the initial *C. funebris* establishment, which has led to intense intra-specific

competition for sunlight and other resources. Consequently, vertical growth is favored over lateral growth in *C. funebris* populations, as previously reported by *Saito, Kawamura & Takeda (2012)* and *Hitsuma et al. (2021)*.

The expansive vertical growth of *C. funebris*, which creates ample growing space and increased light availability, is reflected in annual branches and leaves (*St. Clair, 1994*). As depicted in Fig. S5, this characteristic displays a significant positive correlation between crown height and annual branch length ($p < 0.05$), as well as between tree height and leaf angle ($p < 0.05$). These findings form a basis for selecting and advancing superior varieties of *C. funebris*. Moreover, the findings provide valuable insights for the management of growth in seed orchards and plantation forests.

Principal component analysis (PCA) is an effective method for reducing the number of indicators while maximizing the explanatory power of the original indicator information (*Jolliffe & Cadima, 2016*). This method is commonly employed in the comprehensive evaluation process of seed selection and breeding (*Khadivi-Khub, Sarooghi & Abbasi, 2016*; *Yang et al., 2023b*). During our preliminary investigation, we observed notable differences in the morphological characteristics of *C. funebris* branches within the seed orchard. However, these specific branch morphological variations have not been significantly investigated in previous research studies (*Papageorgiou et al., 2005*; *Yang et al., 2023a*). In this study, we observed significant inter-individual differences in *C. funebris*, particularly in leaf angle, annual branch angle, and annual branch length after dimension reduction (Table S4). The results of the principal component analysis corroborated our observations. Furthermore, the principal component analysis results demonstrated that there was no significant correlation or consistent pattern between each principal component and the provenance of *C. funebris* (Figs. S6D and S6E). Similar results were observed in the clustering analysis, where a total of 180 individual *C. funebris* were categorized into four groups, each comprising individuals from diverse provenances (Fig. S7). This indicated that the clustering results did not significantly correlate with the provenances.

The patterns of trait variation in individual forest trees can be summarized into three types: geographic, continuous, and random (*McKown et al., 2014*; *Suvanto et al., 2016*; *Queiroz et al., 2021*; *Wu et al., 2021b*). The final type, random variation, is less prevalent in related studies. In this study, the phenotypic variation of *C. funebris* exhibited a random pattern that was not geographically related. This pattern was consistent with that observed in *Pinus yunnanensis* (*Liu et al., 2022c*). Apart from the adaptive morphological variations stemming from differences between the original habitat and the current environment (*Singh et al., 2013*), the growth characteristics and reproductive biology of *C. funebris* also contributed significantly to the random variation patterns observed in this species. Due to its extensive distribution and high adaptability, it was often difficult to discern geographical isolation between populations of *C. funebris* (*Jiang & Wang, 1997*). Moreover, *C. funebris*, as a wind-pollinated coniferous species, exhibits a characteristic that creates favorable conditions for gene flow and even facilitates interspecific gene introgression through hybridization (*Petrova et al., 2018*). Consequently, the frequent exchange of genetic material and the presence of adaptive genetic variation serve as primary drivers for the observed random patterns of variation in diverse *C. funebris*

populations. Moreover, according to *Wójkiewicz, Litkowiec & Wachowiak (2016)*, despite the considerable distances between populations of wind-pollinated coniferous trees, as long as they are not isolated, there are no notable genetic differences among them. Based on this, it can be hypothesized that similar patterns of variation are likely to occur in natural populations of coniferous or congeneric trees that exhibit growth characteristics similar to *C. funebris*. This finding has the potential to further our understanding of the geogenetics of these tree species and facilitate conservation and management efforts.

The results of the clustering analysis revealed that the four groups of *C. funebris* exhibited distinct phenotypic differences. Group III exhibited a graceful tree morphology, making it ideally suited for integration into urban greening projects. In contrast, group IV exhibited remarkable growth and seed traits, rendering it an excellent candidate for forestry breeding and genetic enhancement programs. These findings will provide a solid foundation for the classification and utilization of *C. funebris*.

## CONCLUSIONS

The abundant genetic diversity of *C. funebris* serves as a prerequisite for generating variant populations and facilitating continuous selective breeding, thereby enabling the screening of populations with distinct characteristics that hold immense potential for applications in ecological restoration, afforestation, timber production, and urban greening. This study investigated the genetic diversity of *C. funebris* through a comprehensive analysis of phenotypic traits. The results revealed a high degree of genetic variation among 180 germplasm materials sourced from five populations. Cluster analysis identified four distinct groups, each characterized by unique combinations of traits including tree height, wood volume, branch and leaf morphology, cone and seed traits. These findings underscore the richness of genetic resources available for the improvement of *C. funebris*. The identified groups represent valuable genetic pools that can be harnessed for the development of superior cultivars. Future research should focus on elucidating the genetic architecture of key traits and evaluating the performance of these groups across diverse environmental conditions.

## ACKNOWLEDGEMENTS

The authors thank the Santai county national *Cupressus funebris* improvement base, Sichuan Province, China, and its staff for their invaluable support in facilitating this study.

### Funding

This research, conducted by Yu Zhong, was funded by the Innovation of Breakthrough Forest Tree Breeding Materials and Methods and Selection of New Varieties (No: 2021YFYZ0032), and the Key Research and Development Project of Sichuan Science and Technology Plan. The funders had no role in study design, data collection and analysis, decision to publish, or preparation of the manuscript.

## Grant Disclosures

The following grant information was disclosed by the authors:
Innovation of Breakthrough Forest Tree Breeding Materials and Methods and Selection of New Varieties: 2021YFYZ0032.
Key Research and Development Project of Sichuan Science and Technology Plan.

## Competing Interests

The authors declare that they have no competing interests.

## Author Contributions

- Wang Yan conceived and designed the experiments, performed the experiments, analyzed the data, prepared figures and/or tables, authored or reviewed drafts of the article, and approved the final draft.
- Yongqi Xiang conceived and designed the experiments, performed the experiments, prepared figures and/or tables, and approved the final draft.
- Mei Gao performed the experiments, analyzed the data, prepared figures and/or tables, and approved the final draft.
- Ruoyu Deng performed the experiments, analyzed the data, prepared figures and/or tables, and approved the final draft.
- Yan Sun performed the experiments, analyzed the data, prepared figures and/or tables, and approved the final draft.
- Renping Wan performed the experiments, analyzed the data, prepared figures and/or tables, and approved the final draft.
- Xianyi Pan performed the experiments, analyzed the data, prepared figures and/or tables, and approved the final draft.
- Wanzhen Li performed the experiments, analyzed the data, prepared figures and/or tables, and approved the final draft.
- Yu Zhong conceived and designed the experiments, performed the experiments, authored or reviewed drafts of the article, and approved the final draft.

## Data Availability

The raw data is available in the Supplemental File.

## Supplemental Information

Supplemental information for this article can be found online at http://dx.doi.org/10.7717/peerj.18494#supplemental-information.

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
