# Peer review of "Phenotypic diversity and provenance variation of *Cupressus funebris*: a case study in the Sichuan Basin, China"

_PeerJ, doi:10.7717/peerj.18494_

## Round 0.1 · original submission · Minor Revisions

Thank you for submitting your manuscript to PeerJ. We have received and carefully considered the reviews from two expert reviewers. Both reviewers have recognized the significance and quality of your work but have also suggested some minot revisions to improve the clarity and overall impact of the manuscript. Based on the reviewers recommendations, I am pleased to inform you that your manuscript is suitable for publication pending minor revisions. Please address the reviewers comments in a revised version of your manuscript. A detailed response to each comment is also required, explaining how you have addressed the reviewers suggestions or providing a rationale if you choose not to make certain changes.

We appreciate your contribution and look forward to receiving your revised manuscript.

Reviewer 1 ·

Basic reporting

No comment

Experimental design

The study demonstrates an acceptable experimental design.

Validity of the findings

No comments

Additional comments

The study on Cupressus funebris is a valuable contribution to the field of forestry and conservation. By addressing the significant knowledge gap in genetic diversity and morphological characteristics of this species, the researchers have provided crucial insights for its sustainable management and utilization. The findings of substantial phenotypic variation within and between provenances highlight the importance of conserving diverse germplasm resources. The identification of distinct morphological groups with specific characteristics opens up opportunities for targeted breeding programs to develop cultivars with desired traits for different applications.
-Comments and Suggestions for Authors
- Abstract
-The abstract is well-written and informative. It clearly outlines the research problem, the objectives, methodology, key findings, and implications.
- While the abstract states "considerable level of phenotypic variation," providing specific statistical measures (e.g., coefficient of variation, ANOVA results) would strengthen the impact.
- Although the abstract mentions genetic diversity, it would be beneficial to briefly describe the methods used to assess it (e.g., molecular markers, isozymes) or at least indicate that genetic diversity analysis was conducted.
-The statement "the germplasm resources can be categorized into four distinct groups" could be expanded to provide a brief description of the criteria used for grouping.
- Incorporating relevant keywords (e.g., Cupressus funebris, morphological traits, ecological restoration) can improve the abstract's discoverability.
- Introduction
The introduction is comprehensive and well-structured. It effectively establishes the importance of genetic diversity, the role of phenotypic variation, and the specific research gap related to Cupressus funebris. The introduction clearly outlines the objectives of the study.
- Paragraphs 56-66: These paragraphs could be condensed by combining similar points and removing redundant information.
- Paragraphs 97-106: Consider providing more specific details about the phenotypic variations observed in the wild seed collection.
- Materials & Methods
-More details about the specific ImageJ procedures used to measure leaf and seed traits would be helpful. For instance, how were the images calibrated?
- Results
- The results section provides a clear and comprehensive overview of the study's findings.
- Emphasize the statistical significance of findings throughout the text. For example, instead of stating "significant differences," specify the exact p-value (e.g., p < 0.01).
- Line 218: Change "conducted on" to "performed on" for a more concise expression.
- Line 220: Instead of "equally as high as," you can simply say "as high as."
- Line 235: You could replace "illustrates" with "shows" for a more direct style.
- Consider adding the percentage of variance explained by each principal component to provide a more complete picture of the PCA results.
- While the interpretation of principal components is generally clear, you could strengthen it by providing more specific details about the high-loading traits for each component. For instance, instead of saying "cone traits and seed traits," specify which exact traits have high loadings.
- Discussion
- The discussion section effectively summarizes the key findings of the study and places them in the context of existing research.
- Remove the subtitles in discussion section.
- Conclusion
- Briefly explain why the high genetic diversity and distinct groups are important for the species and its potential applications.
- Briefly highlight the most important traits that contributed to the classification of germplasm into different groups.
- Suggest potential future research based on the findings of the study. For example, exploring the genetic basis of the observed phenotypic variation or investigating the performance of the identified groups in different environmental conditions.
Revised Conclusion (Example)
This study investigated the genetic diversity of C. funebris using a comprehensive analysis of phenotypic traits. Results revealed a high degree of genetic variation among 180 germplasm materials from five populations. Cluster analysis identified four distinct groups characterized by specific combinations of growth, branch, leaf, cone, and seed traits. These findings underscore the rich genetic resources available for C. funebris improvement. The identified groups represent valuable genetic pools for developing superior cultivars. Future research should focus on elucidating the genetic architecture of key traits and evaluating the performance of these groups under diverse environmental conditions.

Reviewer 2 ·

Basic reporting

This study conducted a genetic diversity analysis of growth traits, branch and leaf traits, cone, and seed traits in 180 germplasm materials of C. funebris from five different geographical sources. The manuscript is relatively well-written, the experiments are well-performed, and the results are well-presented. However, before publication, several issues need to be addressed, as outlined below

Experimental design

Material
1- Line 138: Could you please provide a table listing the 60 half-sibling families, including the names and parentage for each family
2- Line 143: Table 1 does not correspond to the data mentioned in the text
3- The authors did not mention the seasons during which the study was conducted

Validity of the findings

Results
1- The quality of Figure 6 and 7 need improvement.
2- Line 215: Please provide the analysis of variance for each provenance separately as a supplementary table.
3- Line 211: Please review again. The average coefficient of variation was 13.8%, which I think is not particularly high. Typically, coefficients of variation under 15% are considered acceptable, indicating that the data variability is within a reasonable range and does not suggest significant variations in the measurements.
4- Line 241: please review again
5- Line 236: Could you clarify the benefit of studying the variation among the provenances

Additional comments

Discussion
Please focus on the key findings of this research. The authors should investigate and discuss potential reasons for the observed differences among the 180 C. funebris germplasm resources, despite their selection from half-sibling families.

---

## Round 0.2 · accepted · Accept

I am pleased to inform you that your revised manuscript, having been thoroughly reviewed by the initial two reviewers, has met their satisfaction. Both reviewers have expressed their approval of the revisions made. Based on their positive feedback and the quality of the revisions, I am happy to accept your manuscript for publication.

Congratulations, and thank you for your contribution

Reviewer 1 ·

Basic reporting

no comment

Experimental design

no comment

Validity of the findings

no comment

Additional comments

The authors have made the changes I suggested in the last review. I recommend its publication in this journal.

Reviewer 2 ·

Basic reporting

The authors have made significant progress in addressing most of the suggestions provided in the first review. The improvements made enhance the overall clarity and robustness of the study. However, a few minor grammatical errors and formatting inconsistencies remain. I recommend a final proofread for clarity and consistency.

Experimental design

The experimental design has been improved. The authors have addressed previous concerns.

Validity of the findings

The findings of the study appear to be valid and well-supported by the data presented